# Genes associated with ant social behavior show distinct transcriptional and evolutionary patterns

Alexander S Mikheyev[1,2], Timothy A Linksvayer[3]*

[1]Ecology and Evolution Unit, Okinawa Institute of Science and Technology, Okinawa, Japan; [2]Research School of Biology, Australian National University, Canberrra, Australia; [3]Department of Biology, University of Pennsylvania, Philadelphia, United States

**Abstract** Studies of the genetic basis and evolution of complex social behavior emphasize either conserved or novel genes. To begin to reconcile these perspectives, we studied how the evolutionary conservation of genes associated with social behavior depends on regulatory context, and whether genes associated with social behavior exist in distinct regulatory and evolutionary contexts. We identified modules of co-expressed genes associated with age-based division of labor between nurses and foragers in the ant *Monomorium pharaonis*, and we studied the relationship between molecular evolution, connectivity, and expression. Highly connected and expressed genes were more evolutionarily conserved, as expected. However, compared to the rest of the genome, forager-upregulated genes were much more highly connected and conserved, while nurse-upregulated genes were less connected and more evolutionarily labile. Our results indicate that the genetic architecture of social behavior includes both highly connected and conserved components as well as loosely connected and evolutionarily labile components.

*For correspondence: tlinks@sas.upenn.edu

**Competing interests:** The authors declare that no competing interests exist.

**Reviewing editor**: Philipp Khaitovich, Partner Institute for Computational Biology, China

## Introduction

The main conclusion of a decade of sociogenomic research with a range of solitary and social animal species is that highly conserved genes underpinning core physiological processes can also influence behavioral state (*Amdam et al., 2004*, *2006*; *Toth and Robinson, 2007*; *Toth et al., 2007*, *2010*; *Woodard et al., 2011*; *Woodard et al., 2014*). For example, the insulin signaling pathway, which mediates an organism's response to its internal nutritional state, also influences its behavior (*Ament et al., 2008*). The *genetic toolkit hypothesis* and related hypotheses propose that a conserved set of genes and gene pathways involved in core physiological processes such as metabolism and reproduction has been repeatedly used in the evolution of complex social behavior in diverse lineages (*West-Eberhard, 1996*; *Amdam et al., 2004*, *2006*; *Toth and Robinson, 2007*; *Toth et al., 2007*). This hypothesis stems from findings in Evolutionary Developmental Biology that morphological innovation in disparate lineages often involves the convergent use of a conserved set of genes (e.g., Hox genes) (*Carroll et al., 2001*; *Toth and Robinson, 2007*; *Wilkins, 2013*).

However, social behavior and other social traits are commonly viewed as having unique genetic features and evolutionary dynamics, including especially rapid evolution (*West-Eberhard, 1983*; *Tanaka, 1996*; *Moore et al., 1997*; *Wolf et al., 1999*; *Nonacs, 2011*; *Bailey and Moore, 2012*; *Van Dyken and Wade, 2012*). Could the molecular mechanisms underlying social interactions (e.g., social signal production and response) and social behavior, together with the process of social evolution result in distinct genetic architectures for social traits compared with other traits? Recent comparative transcriptomic and genomic studies find low overlap in genes associated with social behavior in

**eLife digest** Animal species vary widely in their degree of social behavior. Some species live solitarily, and others, such as ants and humans, form large societies. Many researchers have tried to understand the genetic changes underlying the evolution of social behavior. Some researchers suggest that it involves recycling existing genes that also have other conserved functions. Others propose that the evolution of social behavior involves completely new genes that are not found in related but solitary species.

Ants are one of the best-studied social animals. An established colony can contain many 1000s of individuals that live and work together and perform different roles. The queen's job is to lay eggs, while the worker ants do everything else, including collecting food, caring for the young, and protecting the colony. In some species of ant—including the pharaoh ant—a worker's role changes as it ages. Younger workers tend to stay in the nest and nurse the brood, while older workers tend to leave the nest and forage for food.

Mikheyev and Linksvayer asked: which genes are responsible for this age-based division of labor? And how did this aspect of social behavior evolve? First, after observing pharaoh ants from two colonies set up in the laboratory, they confirmed that workers nursing the brood were on average almost a week younger than those seen collecting food. Next Mikheyev and Linksvayer identified which genes were expressed in ants of different ages, or ants engaged in different tasks. Specific sets of genes were expressed more (or 'up-regulated') in nurse workers, while others were up-regulated in foraging workers.

Mikheyev and Linksvayer then investigated how rapidly these genes had evolved by comparing them to related genes found in other social insects (fire ants and honey bees). They also determined the 'connectivity' of these genes by asking how many other genes showed similar expression patterns. In many organisms, how rapidly a gene evolves depends on how tightly connected its expression is to the expression of other genes; highly connected genes evolve more slowly.

The genes that were expressed more in the older foraging workers were both more highly connected and more evolutionarily conserved in the other social insects. Genes that were up-regulated in the younger nurse workers were more loosely connected and rapidly evolving.

Mikheyev and Linksvayer's findings show that the evolution of social behavior in animals involves both new genes, which tend to be loosely connected, and conserved genes, which tend to be more highly connected.

different highly social animals and instead highlight the importance of novel genes and rapid evolution of social traits (*Johnson and Tsutsui, 2011*; *Ferreira et al., 2013*; *Simola et al., 2013*; *Wissler et al., 2013*; *Feldmeyer et al., 2014*; *Harpur et al., 2014*; *Sumner, 2014*; *Jasper et al., 2015*), in accordance with recent studies emphasizing the ubiquity of taxonomically restricted genes (*Domazet-Loso and Tautz, 2003*; *Khalturin et al., 2009*; *Tautz and Domazet-Loso, 2011*). Perhaps social evolution does not consistently use sets of highly conserved genes to the same degree as morphological evolution? The *novel social genes hypothesis* proposes that genes underlying social behavior are often novel socially acting genes or are genes with novel social functions not present in solitary ancestors (*Johnson and Linksvayer, 2010*; *Johnson and Tsutsui, 2011*; *Sumner, 2014*).

Research supporting the genetic toolkit hypothesis has stressed the significant signal of highly conserved genes affecting core physiological processes in transcriptomic data sets for social behavior (*Robinson et al., 2008*; *Toth et al., 2010*; *Fischman et al., 2011*; *Woodard et al., 2011*, 2014; *Toth et al., 2014*). In contrast, research supporting the novel social genes hypothesis has stressed the overall low proportional overlap of genes underlying social behavior in divergent lineages as well as the apparently general low degree of transcriptomic and genomic conservation in divergent lineages (*Johnson and Tsutsui, 2011*; *Ferreira et al., 2013*; *Simola et al., 2013*; *Wissler et al., 2013*; *Feldmeyer et al., 2014*; *Harpur et al., 2014*; *Jasper et al., 2015*; *Sumner, 2014*).

We sought to build on these previous results by examining how transcriptional regulatory context influences evolutionary conservation for genes associated with ant social behavior, and further whether genes associated with ant social behavior exist in distinct regulatory and selective contexts compared to the rest of the genome. Research in a range of model organisms demonstrates that the

degree of a gene's connectivity to the rest of the regulatory network and its level of expression is often negatively correlated with its rate of molecular evolution (*Krylov et al., 2003*; *Hahn and Kern, 2005*; *Jovelin and Phillips, 2009*; *Ramsay et al., 2009*). For example, highly connected 'hub' genes are often highly expressed and evolutionarily conserved. Previous research has compared rates of molecular evolution for genes associated with reproductive division of labor in social insects (*Hunt et al., 2010, 2013*; *Harpur et al., 2014*), as well as other conditionally expressed phenotypes in other organisms (*Brisson and Nuzhdin, 2008*; *Leichty et al., 2012*; *Purandare et al., 2014*), indicating that genes associated with the expression of worker traits experience elevated rates of molecular evolution. However, the relationships among molecular evolution, connectivity, and expression have been little explored in social insects and are generally little understood for genes associated with social behavior. As a result, it is unclear if observed differences in rates of molecular evolution are caused by differences in regulatory architecture, expression, or perhaps result from distinct evolutionary mechanisms such as kin selection, which may operate differentially on genes associated with social behavior relative to the rest of the genome (*Linksvayer and Wade, 2009*; *Hall and Goodisman, 2012*). We further sought to identify modules of co-expressed genes that may be composed of both conserved and novel genes and may contribute to the expression and evolution of social complexity.

We studied the genetic basis and evolution of a fundamental aspect of social insect behavior, age-based division of labor (age polyethism). Age polyethism involves the progression of workers from in-nest tasks such as nursing to outside-nest tasks such as foraging. Because age polyethism is a trait expressed by the functionally sterile worker caste, it is expected to be shaped primarily through kin selection (*Hamilton, 1964*). While age polyethism plays a central role in the functioning of many eusocial systems (*Hölldobler and Wilson, 2009*), the molecular underpinnings have only been well studied in the honey bee *Apis mellifera* (*Whitfield et al., 2006*; *Ament et al., 2008*; *Chandrasekaran et al., 2011*), so that the genetic and evolutionary basis of age polyethism is not generally understood outside of honey bees. We identified transcriptional modules of co-regulated genes associated with worker age polyethism in the pharaoh ant *Monomorium pharaonis*; we identified the degree that these genes overlap with genes involved in age polyethism in two other social insects (*Alaux et al., 2009*; *Manfredini et al., 2014*); and we studied the relationship between expression level, connectivity and rates of molecular evolution at these genes compared to the rest of the genome.

## Results

### Behavioral analysis

We tracked cohorts of age-marked workers and recorded their behavior and location inside and outside the nest. In order to identify differentially expressed genes associated with age-based division of labor, we collected age-marked workers and workers observed performing specific behaviors. The observed location of workers from different age classes changed with both nest location and behavior (glm with quasipoisson errors and log link, both p < 0.01) (*Figure 1*, *Figure 1—figure supplements 1, 2*). In concordance with the expected pattern of age polyethism, the average age of workers observed in the different locations increased as distance from the brood area increased (*Figure 1—figure supplement 2*). Of the 15 behaviors observed more than 15 total times (*Supplementary file 1*), the likelihood of observing workers performing the behaviors 'nurse', 'groom', 'rest', 'trophallaxis', 'walk', and 'forage' depended on worker age (*Figure 1A*; glm with binomial errors and logit link, all nominal p < 0.0002, α = 0.003, controlling for multiple testing). Nursing and foraging were at the two extremes: the average age of workers observed nursing was 6.94 days and the average age of workers observed foraging (i.e., in the act of collecting food) was 13.04 days. There appeared to be a marked transition from nursing to foraging between 9 and 12 days of age (*Figure 1A*), with 75% of nursing observations made for workers less than 10 days old and 75% of foraging observations made for workers over 10 days old (*Figure 1—figure supplement 1*).

### Genome and transcriptome assembly

There was a trade-off in the assemblies between N50 and overall assembly lengths, as a function of kmer size. We chose k = 69 as a compromise between these two metrics, resulting in a scaffolded assembly of 284 mb, with a N50 of 19.0 kb. Although there is no *M. pharaonis* genome size estimate, the assembly is in the range of genome sizes typical of other myrmicine ants (*Tsutsui et al., 2008*). CEGMA analysis (*Parra et al., 2009*) found complete sequences for 92% of the ultra-conserved

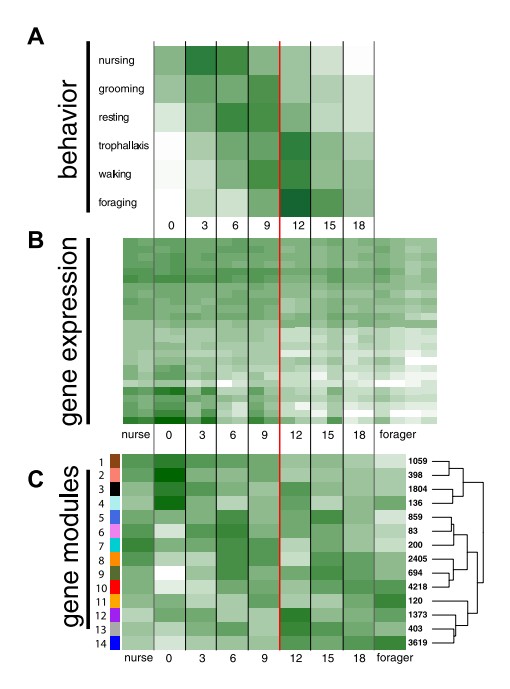

**Figure 1**. Behavioral and transcriptional changes associated with worker age and behavior. Numbers along the x-axis represent ages of marked worker cohorts, starting at worker eclosion as an adult (day 0). In all plots dark green represents greater values, while white represents lower values of the measure being plotted. (**A**) Behavioral results. Workers showed an age-dependent progression of activity, progressing from tasks such as nursing and grooming in the nest to outside tasks such as walking and foraging. (**B**) Heat map of expression levels over the course of worker aging (higher expression in darker green), for 25 genes most differentially expressed between nurses and foragers. The red line separates the samples classified as 'nurses' by K-nearest neighbor classification on the left, from 'foragers' on the right, suggesting a distinct transition between the two categories. (**C**) Correlation between patterns of expression in the 14 identified modules across worker age and behavior. The colors of the boxes are scaled with the value of correlation coefficients, ranging from white to dark green. On the right side of the heat map are the numbers of genes in each module and a dendrogram showing the inferred relationships among modules. The modules show complex patterns of expression, for example with some most highly expressed at age 0, some showing decreasing expression over time, and some increasing expression over time.

The following figure supplements are available for figure 1:

**Figure supplement 1**. The behaviors performed by age-marked workers changed as the workers aged, from nursing to foraging.

*Figure 1. continued on next page*

eukaryotic genes, and partial sequences for 97%. Most reads (97.6%) could be re-mapped to the genome assembly, resulting in a coverage estimate of 40×. Cufflinks assembly identified 22,385 transcribed loci. 74.9 ± 18% (median 85.1%) of the reads for each sample could be re-mapped to predicted transcripts extracted from the reference. After the reads were re-mapped to the assembled transcripts using the RSEM pipeline, each library had 10,602,832 ± 2,925,898 expected counts.

## Gene expression differences associated with worker behavior

The complete analysis of gene expression data, including R code and output, is available in the *Supplementary file 2* (with the complete R markdown script as *Source code 1*), and it is summarized below. We wished to examine which of the four worker behavioral samples (nursing larvae, foraging, grooming larvae, and worker–worker trophallaxis [i.e., exchanging liquid food]) had distinct expression profiles vs all of the others. We used linear contrasts to determine the number of differentially expressed genes between the focal behavioral category and the other behaviors. Of these contrasts, only foragers and nurses had significantly different gene expression patterns, when compared to the rest, that is, there was no evidence that workers engaged in grooming and trophallaxis had distinct transcriptional states. Consequently, we focused subsequent analysis on nurse and forager behavioral categories, except in the construction of the co-expression networks, where all behavioral category and age class samples were used (see below). There were 1217 forager-upregulated, 1247 nurse-upregulated transcripts, and 14,907 transcripts that were not differentially expressed.

## Gene expression associated with age polyethism

Qualitatively, gene expression patterns mirrored the behavioral transition from nursing to foraging that we observed around day 10 (*Figure 1A,B*). To quantify these observations, we used a supervised learning approach (K-nearest neighbor classifier or KNN) to check whether genes differentially expressed in nurses and foragers could be used to differentiate the age class data into two clusters. After the KNN was trained on nurse and forager profiles, it clearly separated workers into two distinct classes based on age, assigning those younger than 12 days into the

*Figure 1. Continued*

**Figure supplement 2**. The location of age-marked workers also changed as the workers aged, from the nest area over the brood to outside the nest.

**Figure supplement 3**. The identified modules vary in expression pattern, composition of nurse-upregulated and forager-upregulated genes, and the proportion of conserved genes with identified fire ant orthologs.

nurse class, and the rest into the forager class (*Supplementary file 2* pages 14–15), suggesting a fairly discrete transcriptomic transition between the two behaviors.

## Gene expression conservation analyses

The proportion of genes with identified orthologs in the fire ant *Solenopsis invicta* differed between behavioral categories (*Manfredini et al., 2014*), with forager-upregulated genes having a higher proportion (0.54) relative to nurse-upregulated (0.43) and non-differentially expressed (0.43) (multiple comparison Kruskal–Wallis, p < 0.05). Similarly, the proportion of genes with identified honey bee *A. mellifera* orthologs was higher for forager-upregulated genes (0.50), relative to nurse-upregulated (0.38), and non-differentially expressed genes (0.38) (multiple comparison Kruskal–Wallis, p < 0.05) (note we used a less conservative BLAST threshold for the honey bee so that the proportions of honey bee and fire ant orthologs are not directly comparable, see 'Materials and methods'). Furthermore, approximately half of non-differentially expressed (0.51) and nurse-upregulated (0.50) genes did not have orthologs identified in either the fire ant or honey bee genomes, but this proportion was lower for forager-upregulated genes (0.39); correspondingly, the proportion of forager-upregulated genes with orthologs identified from both fire ants and honey bees was higher (0.43) compared to nurse-upregulated and non-differentially expressed genes (0.32) ($X^2 = 71.42$, df = 6, p < $10^{-13}$).

Genes previously detected as upregulated in nurses and foragers of *S. invicta* were more likely to have identified *M. pharaonis* orthologs up-regulated in these contexts as well (p = 0.0022 and p = 0.040, respectively). However, the actual percentage of genes differentially expressed in the same context in these two ant data sets was small: 3.8% (47/1247) of nurse genes and 3.2% (39/1217) of forager genes; or if only considering genes with orthologs identified in both species, 8.6% (47/549) nurse genes and 5.9% (39/657) forager genes. While there was low overlap in the lists of differentially expressed genes, there could still be stronger overlap in genome-wide expression profiles when comparing nurse and forager samples between *S. invicta* and *M. pharaonis.* Thus, we estimated the correlation in the change of expression between nurse and forager samples (i.e., log fold change) between the *S. invicta* and *M. pharaonis* datasets for all genes with identifiable homologs. There was a significant correlation in the change of expression for nurse and forager samples, but one that explained only 2% of the variance (Spearman's rho = 0.14, 6324 genes, p < $10^{-16}$).

In contrast to the fire ant and pharaoh ant comparison, previously identified forager- and nurse-upregulated honey bee *A. mellifera* genes (*Alaux et al., 2009*) were not more likely to have *M. pharaonis* orthologs expressed in the same context (p = 0.99, p = 0.98, respectively), consistent with a previous comparison between *S. invicta* and *A. mellifera* (*Manfredini et al., 2014*). The actual overlap in honey bee and pharaoh ant gene lists was higher (71 nurse-upregulated genes and 46 forager-upregulated genes) due to the less conservative BLAST threshold we used for identifying honey bee orthologs, but the honey bee lists were also larger (*Alaux et al., 2009*) and the overlap was not significant.

## Gene ontology analysis

Nurse-upregulated genes were strongly enriched for a range of GO terms associated with metabolism (nearly 50 metabolism-related terms with p < $10^{-5}$; *Supplementary file 3*). Forager-upregulated genes had a more diffuse signal, being relatively more weakly enriched for various GO terms, for example, associated with signal transduction and gland morphogenesis. Forager-upregulated genes showed a more consistent signal for underrepresented terms, for example, GO terms associated with metabolic processes and chromatin modification (*Supplementary file 3*).

## Modules inferred by weighted gene co-expression network analysis (WGCNA)

The number of modules produced by WGCNA can vary based on several thresholding parameters, which we left as defaults (*Supplementary file 2*, pages 20–21). These settings resulted in 14 co-expression modules, ranging in size from 83 to 4218 genes (*Figure 1C*; *Figure 1—figure supplement 3*). A module's

overall expression can be characterized by its eigengene. Correlations between eigengenes and traits in the original data suggest the involvement of corresponding modules in these traits. Eigengenes in two of the modules—1 and 14, which contained the most nurse and forager genes, respectively—were strongly correlated with worker age, although in opposite directions, suggesting their role in aging and age-based division of labor (r = −0.95, r = 0.91 and with FDR-adjusted p-values 0.0038, 0.023, respectively) (*Supplementary file 2*, page 24). Other modules showed complex patterns of age and behavior specific expression, with most of them showing a peak in expression once or twice during the lifetime of a worker (*Supplementary file 2*, page 26). Interestingly, most module eigengenes switched signs during the period between 9 and 12 days, corresponding to the behavioral transition from nursing to foraging. In other words, there appeared to be a major reprogramming step, where modules initially showing low expression became up-regulated, while modules initially showing high expression were down-regulated.

Forager-upregulated genes were concentrated in just a few modules, with only two modules containing more than 100 forager-upregulated genes (*Figure 1—figure supplement 3*). In contrast, nurse-upregulated genes were more widely distributed, with five modules having more than 100 nurse-upregulated genes (*Figure 1—figure supplement 3*). These five modules were mainly enriched for GO terms associated with metabolism and development (*Figure 1—figure supplement 3*; *Supplementary file 4*). Module 5, which contained 116 nurse-upregulated genes, was also enriched for terms associated with female gonad development, which is surprising given that *M. pharaonis* workers lack ovaries and are completely sterile. The modules containing forager-upregulated genes were enriched for a broad range of GO terms, for example associated with regulation of signaling, development and neurogenesis, and gene expression (*Figure 1—figure supplement 3*; *Supplementary file 4*). The proportion of module genes with identified *S. invicta* orthologs ranged from 0.28 to 0.53 (*Figure 1—figure supplement 3*), suggesting that in addition to being involved in different functions, the modules are composed of different proportions of conserved and taxonomically restricted genes.

## Relationship between gene behavioral category, expression level, connectivity, and evolutionary rate

Forager-upregulated genes were much more connected than nurse or non-differentially expressed genes, while nurse-upregulated genes were less connected than non-differentially expressed genes (*Figure 2A*) (multiple comparison Kruskal–Wallis, p < 0.05). There was a small but significant difference in evolutionary rate dN/dS (*Figure 2C*), with nurse-upregulated genes evolving more rapidly than non-differentially expressed genes (multiple comparison Kruskal–Wallis, p < 0.05). Nurse and forager genes were also more highly expressed (*Figure 2B*) than non-differentially expressed genes (Kruskal–Wallis, p < 0.05), although this last comparison is likely biased because differential expression is more easily detected in highly expressed genes.

Co-expression network connectivity and expression level were overall negatively associated with evolutionary rate, such that highly connected and highly expressed genes had decreased rates of molecular evolution (*Figure 2D,E*; evolutionary rate and connectivity, r = −0.15, p < 2 × 10⁻¹⁶; evolutionary rate and expression, measured in terms of transcriptional abundance, fragments per million reads mapped, FPKM, r = −0.12, p < 2 × 10⁻¹⁶); and connectivity and expression were positively correlated (r = 0.30, p < 2 × 10⁻¹⁶). In a full model considering how a gene's rate of molecular evolution depended on its gene expression level, network connectedness, and behavioral category, the largest effects were main effects of expression (z = −7.42, p = 1.29 × 10⁻¹³) and connectivity (z = −3.69, p = 0.00023).

We also studied the effects of gene category (i.e., upregulated in nurses or foragers, or not differentially expressed), expression level, and connectivity on whether a given *M. pharaonis* gene had an identifiable fire ant *S. invicta* and honey bee *A. mellifera* orthologs. Overall, genes with orthologs in the fire ant or honey bee had greater connectivity and expression (*Figure 3*, *Figure 3—figure supplement 1*). In considering a model with both main and interaction effects of behavioral category, expression level, and connectivity, connectivity had the strongest effect (glm with quasibinomial residuals: t = 24.5, p < 10⁻¹⁶, for the presence of *S. invicta* orthologs; t = 32.2, p < 10⁻¹⁶, for the presence of *A. mellifera* orthologs), with more highly connected genes being more likely to have an ortholog. There were also much smaller interaction effects indicating that nurse-upregulated genes had fewer orthologs than expected given their connectivities (i.e., connectivity had a weaker effect on nurse-upregulated genes than other genes, *Figure 3* and *Figure 3—figure supplement 1*; t = −3.17,

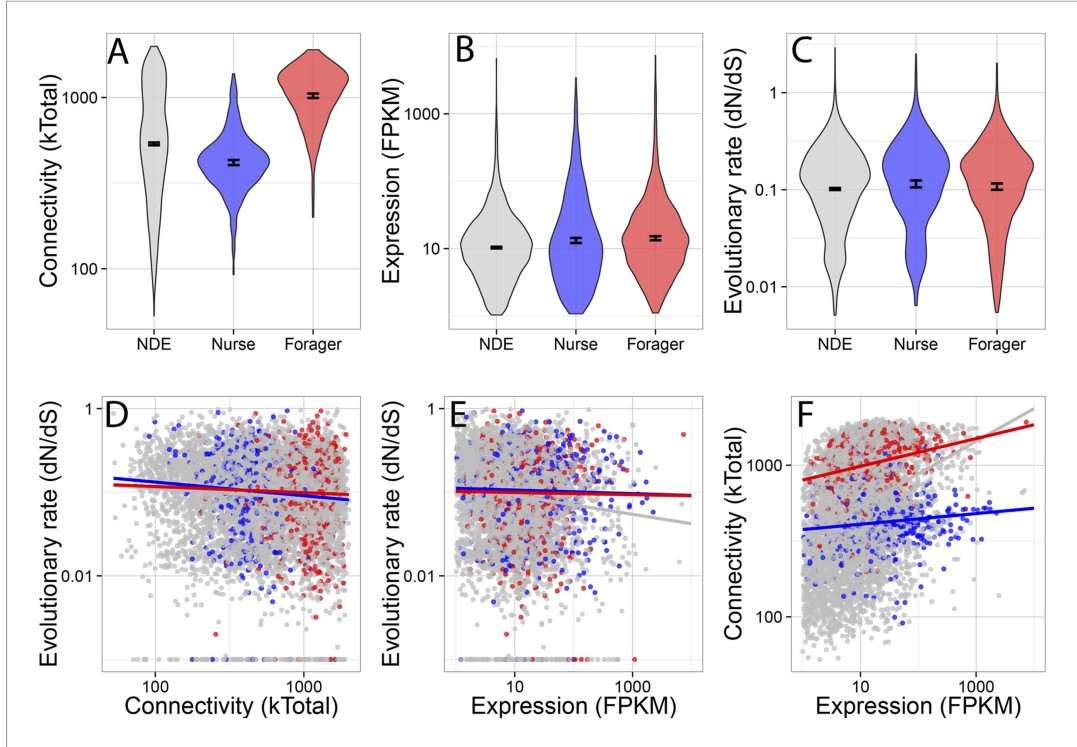

**Figure 2**. Connectivity, expression, and evolutionary rate for nurse-upregulated (blue), forager-upregulated (red), and non-differentially expressed genes (gray). Overall, connectivity and expression are positively correlated (**F**) and negatively associated with evolutionary rate (**D** and **E**), as expected. At the same time, forager-upregulated genes are much more strongly connected while nurse-upregulated genes are more loosely connected compared to non-differentially expressed genes (**A**); Nurse-upregulated genes have a small but significant increase in evolutionary rate (**C**); and both forager- and nurse-upregulated genes are more highly expressed than non-differentially expressed genes (**B**). The top panels show results for all data, while the bottom panels show results only for genes with *S. invicta* orthologs that had estimated evolutionary rates.

p = 0.0015 for *S. invicta* orthologs; t = −2.76, p = 0.0057 for *A. mellifera* orthologs), and forager-upregulated genes had fewer orthologs than expected given their expression (t = −2.33, p = 0.02 for *S. invicta* orthologs; t = −2.58, p = 0.0098 for *A. mellifera* orthologs; *Figure 3* and *Figure 3—figure supplement 1*).

## Discussion

Pharaoh ant workers showed a clearly defined age-based transition from nursing to foraging, in terms of both behavioral and transcriptional patterns, with nurses and foragers having strongly differentiated sets of upregulated genes (*Figure 1*). We recovered the commonly observed genome-wide relationship between a gene's rate of molecular evolution, its expression level, and its network connectivity (*Krylov et al., 2003*; *Hahn and Kern, 2005*; *Jovelin and Phillips, 2009*; *Ramsay et al., 2009*). Specifically, the rate of molecular evolution (dN/dS) as well as the likelihood a gene had identified fire ant and honey bee orthologs was negatively correlated with its expression level and connectivity within co-expression networks, while expression and connectivity were positively correlated (*Figures 2, 3*). In addition to these genome-wide patterns, nurse- and forager-upregulated genes had distinct regulatory and evolutionary patterns relative to each other and to the rest of the transcriptome (*Figures 2, 3*). Most strikingly, forager-upregulated genes were much more highly connected and correspondingly more conserved, while nurse-upregulated genes were less connected, and more rapidly evolving and less conserved.

Previous studies of the evolutionary genetic basis of social behavior have focused on the overlap of genes lists associated with social traits in different lineages. We found significant but seemingly low

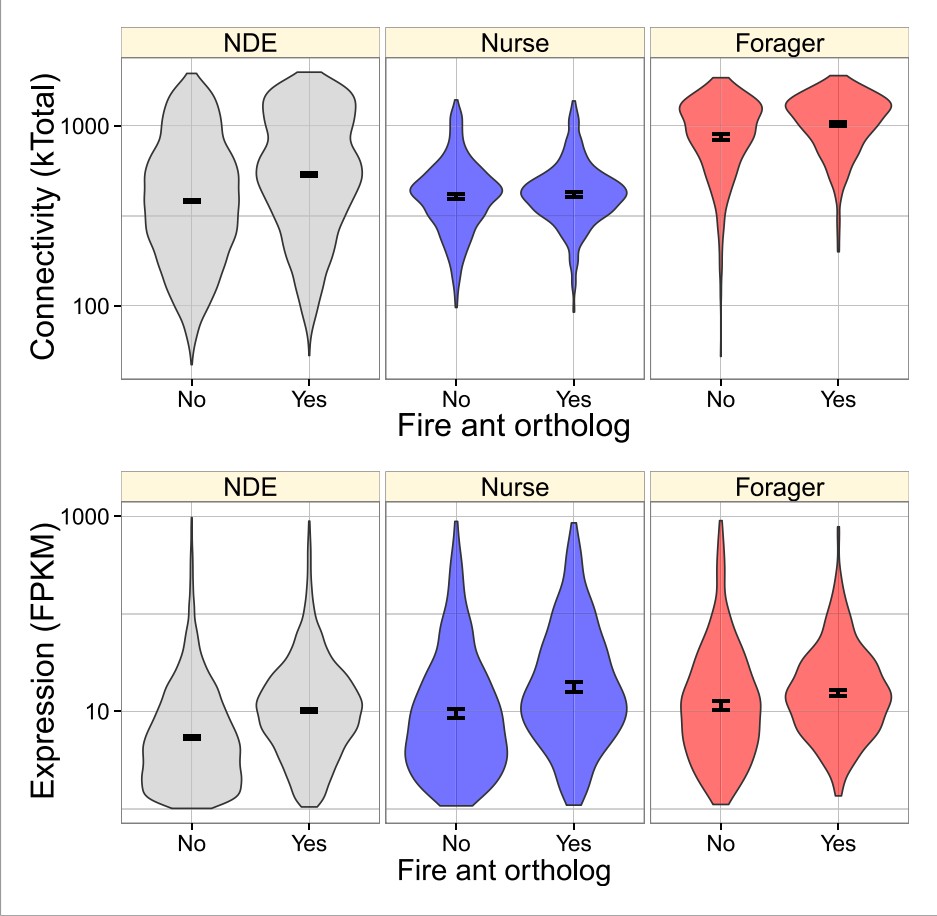

**Figure 3**. Genes with identified fire ant orthologs were more highly connected and expressed, but this relationship also depended on whether the gene was nurse-upregulated (blue), forager-upregulated (red), or non-differentially expressed (NDE, gray). As shown in **Figure 2**, forager-regulated genes were much more highly connected, and overall, forager-upregulated genes had a higher proportion of identified fire ant orthologs (0.54) relative to nurse-upregulated and non-differentially expressed genes (0.43).
The following figure supplement is available for figure 3:

**Figure supplement 1**. Very similarly to **Figure 3**, genes with identified honey orthologs were more highly connected and expressed, but this relationship also depended on whether the gene was nurse-upregulated (blue), forager-upregulated (red), or non-differentially expressed (NDE, gray).

(<4%) overlap in lists of differentially expressed genes and the correlation in genome-wide expression profiles (r = 0.14) when comparing gene expression in nurse and forager samples between the pharaoh ant and fire ant, *S. invicta*. Such low overlap seems surprising, given that these two ants are in closely related ant genera, having diverged on the order of 50 Mya (*Ward et al., 2014*). However, the comparison is not perfect, given substantial differences between the two studies in methodology used to characterize the behaviors, and in the technology used to measure gene expression (i.e., microarray vs RNA sequencing) (*Manfredini et al., 2014*). We did not find significant overlap between lists of honey bee and pharaoh ant genes associated with age polyethism, consistent with results reported by the earlier fire ant study (*Manfredini et al., 2014*). While we expected decreased overlap given that honey bees and ants diverged longer ago, ~170 Mya (*Ronquist et al., 2012*), and represent independent origins of eusociality, the ant-honey bee comparison is also more problematic because the honey bee data are based on brain gene expression profiles whereas the fire ant and pharaoh ant data are based on whole body gene expression profiles.

Past studies have often interpreted significant but similarly low overlap in lists of genes associated with social behavior from different lineages as supporting the genetic toolkit hypothesis

(*Toth et al., 2010*, *2014*; *Woodard et al., 2014*). In contrast, other authors have recently interpreted low overlap as being consistent with the novel social genes hypothesis, which emphasizes the importance of taxonomically restricted genes (*Ferreira et al., 2013*; *Feldmeyer et al., 2014*; *Sumner, 2014*). The contrasting emphasis of authors on either conserved or novel genes begs the question: what degree of conservation in gene lists is necessary for confirmation or rejection of these two hypotheses? For example, the fact that nurse-upregulated genes in *M. pharaonis* are more rapidly evolving than the rest of the genome and that 50% of nurse-upregulated genes do not have identifiable fire ant or honey bee orthologs suggests that novel genes may have important nurse-specific functions. At the same time, the significant overlap of fire ant and pharaoh ant gene lists and the strong enrichment of nurse-upregulated genes for gene ontology terms associated with metabolism and development suggests that conserved genes involved in core physiological processes also play important roles in nurse function and the evolution of division of labor. Thus, our results are generally consistent with both hypotheses. We suggest that neither of these two hypotheses has yet been formulated in a way that is readily tested, in part because it is unclear what precise genes are expected to be included or excluded from a genetic toolkit (*Wilkins, 2013*). Furthermore, these hypotheses are not mutually exclusive, since both conserved and novel genes likely play roles in the evolution of all new traits (*Johnson and Linksvayer, 2010*; *Woodard et al., 2011*).

We suggest that shifting the focus, from lists of genes to modules of co-expressed genes in the context of genome-wide transcriptional and evolutionary patterns, can help to elucidate how social evolution has produced social complexity. In this way, one question we can ask is whether we see any simple molecular signature of social evolution, for example due to kin selection? As *Monomorium* ant workers are obligately sterile, all worker traits are expected to be shaped exclusively by indirect selection (i.e., kin selection) (*Hamilton, 1964*). All-else-equal, such indirect selection is weaker than direct selection, proportional to relatedness (*Hamilton, 1964*), and a priori is expected to produce relaxed selective constraint and elevated rates of molecular evolution for all genes associated with worker traits (*Linksvayer and Wade, 2009*). Past studies have found different rates of molecular evolution for worker-biased and queen-biased genes, with most studies finding that worker-biased genes are more rapidly evolving (*Ferreira et al., 2013*; *Feldmeyer et al., 2014*; *Harpur et al., 2014*; but see; *Hunt et al., 2010*). Some researchers have interpreted different patterns between lineages as being consistent with simple kin selection predictions based on differences in within-colony relatedness (*Hall and Goodisman, 2012*), but most studies have emphasized the association between conditional expression and relaxed selection (*Hunt et al., 2011*, *2013*), as well as genes associated with worker traits simply experiencing stronger positive selection (*Hunt et al., 2010*; *Ferreira et al., 2013*; *Feldmeyer et al., 2014*; *Harpur et al., 2014*). We observed weakly elevated rates of molecular evolution at nurse-upregulated genes compared to the rest of the genome, but much more notable was the distinct connectivity and corresponding differences in gene conservation for forager-upregulated genes relative to nurse-upregulated and non-differentially expressed genes. These results suggest that social evolution does not just have simple genome-wide effects such as relaxed effective selection associated with kin selection, but instead shapes complex social traits while acting within general systems-level constraints imposed by regulatory architecture.

The common perception that social evolution often involves rapid evolutionary dynamics (*West-Eberhard, 1983*; *Tanaka, 1996*; *Moore et al., 1997*; *Wolf et al., 1999*; *Nonacs, 2011*; *Bailey and Moore, 2012*; *Van Dyken and Wade, 2012*) may result from the fact that genes influencing many key social traits are not only conditionally expressed, but are also located peripherally within regulatory networks, and so are relatively unconstrained. For example, we expect that traits associated with social signal production (e.g., pheromone and glandular secretions) are often located peripherally within regulatory networks and as a result may be evolutionarily labile (*Jasper et al., 2015*), as is the case more generally with secreted proteins (*Julenius and Pedersen, 2006*; *Liao et al., 2010*; *Nogueira et al., 2012*). More core and conserved components are also certain to be important to the expression of these traits, but their contribution to trait evolution may be minimized by virtue of the fact that they are highly connected. These arguments suggest how both conserved, toolkit genes, as well as rapidly evolving and taxonomically restricted novel genes, likely play important roles in the evolution of social novelty, with novel genes being added peripherally to regulatory networks. Our results are consistent with this interpretation, because *M. pharaonis* age-based division of labor seems to have a complex genetic basis with some components that are highly

connected and conserved, and other components that are more loosely connected and evolutionarily labile.

Our findings that nurse-upregulated genes are more rapidly evolving and less conserved among social insect lineages relative to forager-upregulated genes suggest that nurse traits have been a major focus of evolutionary innovation between social insect lineages. This result seems surprising given that foragers of different lineages experience diverse environments outside the nest compared to the relatively constant within-nest environment experienced by nurses and could be expected to experience more diverse selective pressures. One explanation is that the physiological mechanisms associated with metabolically costly foraging activities and older adult life (*M. pharaonis* workers usually only live several weeks [*Peacock and Baxter, 1950*], so that foragers which start right before their second week of age may already be senescing) may be relatively conserved and simple. Nursing behavior, occurring during very early adult life, may involve more diverse physiological and developmental processes, and nursing itself may also involve more diverse behaviors and physiological processes, such as food processing and the synthesis of glandular secretions that are fed to larvae. Perhaps the relatively more complex genetic architecture (less tightly connected, involving more modules, and diverse processes) has served as less of a constraint and facilitated more evolutionary change for nurse-related genes. If so, we predict that nurse-specific functions and functions for early adult life may be generally more evolutionarily labile as well as more physiologically and behaviorally labile within and across lineages than forager-specific functions. Note that this prediction is opposite the typical expectation that genes acting early in development have more pleiotropic effects and are thus especially constrained (*Roux and Robinson-Rechavi, 2008*; *Piasecka et al., 2013*), but obligate sterility may, in part, release workers from these constraints on the evolution of genes acting early in worker development.

We identified two discrete sets of genes with distinct genetic architecture associated with age-based division of labor. The majority of forager-upregulated genes were contained within a single gene module (module 14; *Figure 1—figure supplement 3*) that was significantly positively associated with age. Another module with expression negatively associated with age contained the largest number of nurse genes, but nurse genes were also broadly spread out across a number of other modules with complex expression patterns across age and behavioral groups. Interestingly, the modules differed in the proportion of constituent genes which had identifiable *S. invicta* and *A. mellifera* orthologs, indicating that modules vary in the degree to which they are composed of conserved genes and gene networks vs rapidly evolving genes with unknown function. That said, the modules were enriched for various gene ontology terms, providing some insight into their putative functional importance (*Supplementary file 4*).

By explicitly studying regulatory architecture and inferring modules of tightly connected genes in other species as well as *M. pharaonis*, it will be possible to further identify what network components contribute to the expression of social traits, how rapidly these components are evolving within populations, and how they have contributed to phenotypic differences between divergent lineages. Building on the genetic toolkit conceptual framework, it will be possible to ask to what degree diverse lineages repeatedly use the same modules, and importantly approaches already exist for quantifying module overlap in the absence of functional information (*Oldham et al., 2006*; *Langfelder et al., 2011*). Similarly, after finding non-significant overlap in lists of genes associated with queen- and worker-caste development in paper wasps and honey bees, *Berens et al. (2014)* recently invoked a 'looser' version of the genetic toolkit hypothesis by examining the overlap of inferred functional enrichment of gene lists (i.e., via gene ontology analysis). Focus on co-expressed modules may actually improve the feasibility of inferring the function of co-expressed genes based on observed expression patterns together with standard functional information inferred from the subset of conserved annotated genes with identifiable orthologs from model systems. It will also be possible to determine the relative contribution of conserved vs taxonomically restricted genes to co-expression modules.

## Materials and methods

### Colony setup

Two replicate *M. pharaonis* observation colonies were established, each with 10 queens, approximately 4000 workers, and 1000 brood, representing a random subsample of a larger source

colony. Each colony was established from a separate source colony, which came from a stock of approximately 40 colonies that have been repeatedly mixed across generations so that they are genetically similar. Observation nests were constructed of two pieces of 5 × 15 cm glass separated by 1.5 mm thick plastic sheeting. Colonies were given water in cotton-plugged test tubes, 50% honey solution, beef liver, egg yolk, and mealworms *ad libitum*, replaced twice a week. Colonies were maintained at 27 C and 65% relative humidity in climate controlled rooms at the University of Pennsylvania.

Every 3 days, 600 newly eclosed callow workers, which were inferred to be approximately 0–1 days old, were collected from 8–10 stock colonies. These callow workers were briefly anesthetized with $CO_2$ and individually paint marked on the gaster with a unique age cohort color dot using a Sharpie extra fine oil based paint pen, and then 300 were added to each of the observation colonies. Five uniquely marked age cohorts were thus added to the colonies on days 1, 4, 7, 10, and 13 of the study. Nestmate recognition is at most weak and transient in *M. pharaonis* (*Schmidt et al., 2010*), and callows in particular are readily accepted. We also set up a camera to automatically take images of the nest areas of each colony once every 20 min for the entire period of the study, although we do not further discuss these images.

Previous literature indicates that *M. pharaonis* workers are expected to live 9–10 weeks (*Peacock and Baxter, 1950*), but our preliminary trials with our setup indicated that workers tend to die or lose their paint marks after several weeks. We ran the study for 1 month, expecting to capture the major age-based transitions in worker behavior (e.g., the nursing to foraging transition observed in other species), but it is possible that we missed late behavioral transitions that occurred towards the end of workers' lives. In practice, such late transitions are difficult to detect as sample size necessarily declines as increasing numbers of workers die.

## Behavioral analyses

A behavioral scan of each colony was completed once each day for the duration of the month-long study by recording the instantaneous behavior and location observed for every visible paint-marked worker. Each behavioral scan was performed at 20× magnification with a Nikon SMZ800 stereomicroscope. We recorded 30 distinct behaviors, but only 15 were observed more than 15 total times during the study period (*Supplementary file 1*). We defined an individual as foraging if it was observed on a food or water source or actually carrying food (i.e., foraging included the behaviors 'on honey', 'on liver', 'on water', or 'carrying food'; *Supplementary file 1*). Each experimental colony contained four identifiable locations that were redefined prior to each behavioral scan: brood area, brood periphery, remaining nest area, and foraging area. The brood area was defined as the central area within the nest containing all brood and queens (*Edwards, 1991*). The nest periphery was defined as the region directly adjacent to the brood area, where workers were dense in aggregation but not in contact with any of the brood. The nest area was defined as the sparsely occupied remainder of the space within the nest, not including the brood area and nest periphery. The foraging area included all areas outside of the nest. Analyses of behavioral data were conducted in R (www.r-project.org).

## Worker sampling, genomic DNA sequencing, mRNA amplification, and RNA library preparation

Every 3 days, whole bodies of five individuals from each available uniquely paint marked age cohort were collected, flash frozen in liquid nitrogen, and stored at −80°C. This sampling scheme resulted in seven groups of individuals of known age (0, 3, 6, 9, 12, 15, and 18+ days old). 20 individuals of each of these age category were pooled for whole body RNA extraction for each of the two replicate observation colonies. In addition, for each of the two replicate observation colonies, we collected and pooled 20 non-paint marked workers in the act of the following five behaviors: nursing larvae, grooming larvae, engaged in trophallaxis with other workers, foraging for protein (collecting egg, mealworm, or liver), and foraging for carbohydrates (collecting honey solution). RNA was extracted from pools of worker samples of known age or observed behavior using Qiagen RNeasy kits with standard protocols. RNA sequencing libraries were constructed at the University of Arizona Genetics Core (UAGC) with RNA TruSeq library construction kits following standard protocols. In total there were 24 libraries: 2 colony replicates × (7 age groups + 5 behavioral groups). RNA sequencing was conducted at the University of Arizona Genetics Core on an Illumina HiSeq2000 with 100 bp paired ends reads, with six samples multiplexed per lane, randomly distributed across four lanes.

Sequences were post-processed by cutadapt (*Martin, 2011*) to remove Illumina adapter sequences and ConDeTri (*Smeds and Künstner, 2011*) to remove low-quality bases.

## Reference genome sequencing and assembly

DNA from a single haploid male (183 ng) was used to prepare a TruSeq library, which was sequenced in multiplex on an Illumina HiSeq 2000, yielding 70,894,179 million 100 bp read pairs. Raw genomic reads were quality and adaptor trimmed using ConDeTri and cutadapt (*Martin, 2011*; *Smeds and Künstner, 2011*), producing 57,002,951 read pairs and 8,361,560 single reads (12.3 Gb total). The assembly was carried out using ABYSS, with a range of kmers from 53 to 91 (*Simpson et al., 2009*). We then chose the assembly with the longest N50 as the reference for transcriptome assembly. Genome assembly quality was evaluated using the CEGMA pipeline (*Parra et al., 2009*), and by re-mapping the paired end trimmed reads using bowtie2 (*Langmead and Salzberg, 2012*).

## Reference-based transcriptome assembly, annotation and differential gene expression analysis

The transcriptome was mapped to the reference using Tophat 2, and assembled into transcripts using Cufflinks 2.1 (*Roberts et al., 2011*; *Kim et al., 2013*). Gene expression data were obtained by re-mapping the transcript reads to the extracted transcripts using RSEM and calculating the expected counts at the gene level (*Li and Dewey, 2011*). When multiple isoforms of a single locus were found, only the longest transcript was used for gene annotation. Assembled transcripts were annotated using BLASTX from the non-redundant NCBI database with expectation values of $E = 10^{-5}$. These results were used to assign Gene Ontology (GO) profiles with *Blast2go* (*Conesa et al., 2005*).

## Differential gene expression analysis and transcriptional network analysis

Transcript counts were filtered by abundance, removing those with less than 1 fragment per kilobase mapped (FPKM) in more than half of the libraries (*Mortazavi et al., 2008*). Differential gene expression analysis was carried out in edgeR, using a GLM fit to the count data and identifying differentially expressed genes using planned linear contrasts (*Robinson et al., 2010*). In order to infer co-expression modules and gain an insight into network structure of gene interactions, we performed a weighted gene co-expression network analysis (WGCNA) on the count data (*Langfelder and Horvath, 2008*). WGCNA was conducted on the entire transcript set, after filtering out the low-abundance transcripts. This analysis relies on patterns of gene co-expression, but has been shown to reconstruct protein–protein interaction networks with reasonable accuracy (*Zhao et al., 2010*; *Allen et al., 2012*). We used total connectivity as a measure of gene interaction strength, because it is not as sensitive to module assignments, and most likely reflects the overall selective pressures acting on the gene, beyond those imposed by its role in age polyethism. As with most gene expression analysis, WGCNA provides better estimates for highly abundant genes, and in particular for genes showing variation in their expression levels. Consequently, low-abundance and invariant genes will show lower connectivity.

GO term enrichment analysis was performed using the R package GOstats (*Falcon and Gentleman, 2007*). We report GO terms as enriched when p < 0.05.

## Evolutionary rate and gene expression conservation analyses

Fire ant (*S. invicta*) orthologs for each gene were determined using reciprocal best BLASTP, using cutoffs of $10^{-10}$. This parameterization allowed for high specificity, though at the cost of sensitivity, since paralogs were ignored (*Chen et al., 2007*). These results were used to predict the *M. pharaonis* coding sequence using ORFPredictor (*Min et al., 2005*). We then computed the pairwise dN/dS ratios for each gene using the branch model in PAML (v. 4.6). Using the list of differentially expressed genes in foragers vs nest workers in the fire ant (*Manfredini et al., 2014*), Fisher's exact tests were used to examine whether genes differentially expressed in these categories of workers were more likely conserved, than expected by chance. We repeated the analysis above using honey bee (*A. mellifera*) genes, except that the BLAST cutoff was lowered to $10^{-5}$ to increase the chance of identifying orthologs in the more divergent honey bee.

To initially study whether evolutionary rate (dN/dS), connectivity (kTotal), and expression (FPKM) differed between behavioral categories (nurse-upregulated, forager-upregulated, and non-differentially expressed), we used a Kruskal–Wallis test, adjusted for multiple comparisons (kruskalmc function in the R package pgirmess). Finally, to study the main and interaction effects of connectivity, expression, and behavioral category on evolutionary rate, we used a linear model log transformed rate as the dependent variable, log transformed connectivity and expression as continuous predictors, and behavioral category as a categorical predictor.

## Statistical analysis

Statistical analysis was performed with R. Means are presented $\pm$ their standard deviations. p-value cutoffs of 0.05 were used throughout the analysis. In the case of differential gene expression, data analyses were corrected for multiple comparisons using the Benjamini-Hochberg (FDR) procedure (*Benjamini and Hochberg, 1995*).

## Code and data availability

Scripts for the bioinformatic analyses, and a README explaining the workflow can be found at https://github.com/mikheyev/monomorium-polyethism. Most of the workflow and output is shown in *Supplementary file 2*, with the corresponding R script shown in *Source code 1*. All behavioral and gene expression data, including a MySQL database for the gene expression data have been deposited to Dryad, doi:10.5061/dryad.cv0q3 (*Mikheyev and Linksvayer, 2014*). Raw reads and the genome assembly are available at the DNA Data Bank of Japan, DDBJ BioProject PRJDB3164.

## Acknowledgements

Sandra Rehan and Nadeesha Perera collected the behavioral observation data and did RNA extractions. We thank Fabio Manfredini for help with the fire ant data. We are grateful to Yannick Wurm, Eyal Privman, and Laurent Keller for providing the raw *M. pharaonis* genomic reads and to Yannick Wurm for bioinformatic assistance early on in the project. This project was funded in part by a University of Pennsylvania University Research Foundation grant to TAL.

## Additional information

### Funding

| Funder | Grant reference number | Author |
|---|---|---|
| University of Pennsylvania | University Research Foundation grant | Timothy A Linksvayer |

The funder had no role in study design, data collection and interpretation, or the decision to submit the work for publication.

### Author contributions

ASM, Acquisition of data, Analysis and interpretation of data, Drafting or revising the article; TAL, Conception and design, Acquisition of data, Analysis and interpretation of data, Drafting or revising the article

## Additional files

### Supplementary files

• Supplementary file 1. Table with a list of behaviors observed during the study; behavioral code, name of the behavior, description of the behavior, the label for the behavior used in the main text are shown, and the total number of observations for each behavior are shown.

• Supplementary file 2. A PDF of the full R script and abbreviated output with the analyses included in the study.

• Supplementary file 3. A table showing the results of Gene Ontology enrichment analysis for the set of genes upregulated in foragers and nurses. The columns show: the GO Biological Process ID, the p value, Odds Ratio, Expected Count, Observed Count, and total number of genes in the data set matching the GO term, a description of the GO term, whether the term was enriched for forager-upregulated genes or nurse-upregulated genes, and finally whether the term is over- or underrepresented.

• Supplementary file 4. A table showing the results of Gene Ontology enrichment analysis for genes within each of the 14 identified co-expression modules. The columns show: the GO Biological Process ID, the p value, Odds Ratio, Expected Count, Observed Count, and total number of genes in the data set matching the GO term, a description of the GO term, and the module number and module color in which the GO term was overrepresented.

• Source code 1. Complete R Markdown script used in the analyses and to produce *Supplementary file 2*.

### Major datasets

The following datasets were generated:

| Author(s) | Year | Dataset title | Dataset ID and/or URL | Database, license, and accessibility information |
|---|---|---|---|---|
| Mikheyev AS, Linksvayer TA | 2014 | Data from: Genes associated with ant social behavior show distinct transcriptional and evolutionary patterns | http://datadryad.org/resource/doi:10.5061/dryad.cv0q3 | Available at Dryad Digital Repository under a CC0 Public Domain Dedication. |
| Mikheyev AS, Linksvayer TA | 2014 | Data from: Genes associated with ant social behavior show distinct transcriptional and evolutionary patterns | http://trace.ddbj.nig.ac.jp/BPSearch/bioproject?acc=PRJDB3164 | Publicly available at the DDBJ BioProject. |

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
