## [Decision Letter]

Thank you for sending your work entitled “Genes associated with ant social
behavior show distinct transcriptional and evolutionary patterns” for
consideration at *eLife*. Your article has been favorably evaluated by
Diethard Tautz (Senior editor), a Reviewing Editor, and three reviewers.

The Reviewing editor and the reviewers discussed their comments before we reached this
decision, and the Reviewing editor has assembled the following comments to help you
prepare a revised submission.

While all three reviewers are enthusiastic about the study, several important concerns
were voiced by all of them. All reviewers pointed out that this paper does not provide
the necessary statistical details to be able to assess the quality of the work. This is
certainly the most important concern, as hypothesis acceptance/rejection—which is
the central message of the study—fully depends on gene expression analysis and
its interpretation. The respective comments are combined below and it will be necessary
to address them before a final decision about acceptance can be reached.

More specifically:

The authors investigate the “novel social genes” vs the “gene
toolkit” hypotheses. This is certainly interesting and worthwhile. However, one
general issue I have with this (and most other) studies on the topic is that it is not
clear what “exactly” constitutes evidence for one hypothesis or the
other.

For example, the authors here state that the percentage of genes differentially
expressed in the same context in the two ants is small (∼3%). However, the
overlap is marginally significant (fifth paragraph of the Results section). So the
question is, what percent overlap exactly would constitute support for the toolkit
hypothesis? 50%? 25%? 10% 5%?

The point is that it isn't clear how and when authors reject or fail to reject
the toolkit/novel hypotheses because there is never an explicit significance or overlap
threshold provided. The authors are not alone in having to deal with this issue, and so
I don't want to lay this completely at their feet, but it would be nice if they
stated explicitly what exactly would constitute evidence, or lack thereof, for the
toolkit hypothesis somewhere before they get to the results.

Related to this issue, I wanted more detail about the analyses comparing expression
patterns between taxa. Did the authors just ask if the same genes were differentially
expressed between behavioral types in the two taxa to be viewed as
'consistent' with the toolkit model? Or did a particular gene have to be
differentially expressed in the same direction in both taxa? (The latter, I think,
although this wasn't clear.) And did a gene need to be significantly
differentially expressed to be counted? Or was it sufficient that it show the same
directionality in expression, regardless of significance? Directionality without
significance is as important as significance, given that the studies in the taxa had
different power, used different methods, etc. These are not trivial issues and they may
affect the outcome and interpretation of the results. I urge the authors to look into
this more closely.

In the third paragraph: The authors state they are interested in the “molecular
mechanisms of social interactions (e.g. social signal production, reception and
response)”. They refer to genes related to social behavior throughout the
manuscript. But are their expression profiles indeed reflecting social information
producing/processing skills? Or morphological changes related to other functions, such
as exposure to exterior environments? They do not state they used whole ants or just ant
heads for transcriptome profiling, which is highly crucial for interpreting the results
(especially given that some other studies have used the animal head).

In the third paragraph of the Results section: “Of these contrasts, only foragers
and nurses had significantly different gene expression patterns.” This is not
well explained. Four categories are compared and 2 of these are said to have different
expression. What is the reference here; are the other two categories (grooming and
trophallaxis) not different from these two? Or perhaps I am missing something?

This can be partly followed in the Supplementary Material, but should be referred in the
main text, e.g. it seems as if all samples were grouped together in the DE analysis. The
foragers and nurses were most different as they represent the youngest and oldest. I
would have stated this explicitly.

I am also concerned somewhat about the PCA in the Supplementary Material: There seems to
be two groups emerging, but this is likely technical (I would guess sample processing
dates). It might be difficult to control for this, but if possible, could improve DE
analysis significantly.

In the third paragraph of the Results section: “There were 1217 forager- and 1247
nurse-upregulated genes”. What was the p-value cutoff? How did the authors
control for multiple-testing? (This can also be followed in the Supplementary Material,
but should be referred in the main text.)

In the fourth paragraph of the Results section: “(…) it separated workers
into two distinct classes based on age”. If I understand what was done, I think
the authors might be overinterpreting: the algorithm will separate the profiles into 2
classes if k=2, and n classes if k=n. Thus, without additional analysis I
think one cannot decide on the existence of distinct classes. The authors could consider
applying some other test; e.g. check the slope of the expression-age curve.

In the fifth and sixth paragraphs of the Results section: In the gene expression
conservation analysis, we are given no information how many genes are used in the
comparisons (i.e, the number of genes showing DE in both this and the other datasets, as
well as background genes). If the numbers are low, they could instead check the effect
size of orthologous genes identified as DE for honeybee, for example. Was the honeybee
data generated by Manfredini et al., 2014? If not, the authors should state that.

Most importantly, if the honeybee data was generated from the brain (as done by quite a
few studies) and the data in this study from the whole body, this could also be a reason
for finding limited overlap.

In comparisons with the Fisher's exact test, it would be useful to state what the
background is (non-DE genes, genes up-regulated in the other category, or both?).

The expression “whether genes differentially expressed in these categories of
workers were more likely conserved” is a bit confusing, as it also implies
sequence conservation, but I think the authors mean conservation with respect to
correlated changes.

In the seventh paragraph of the Results section: Connectivity—this could be more
explicitly defined, such as emphasizing that the prediction comes from transcription
data correlations (e.g. not protein-protein interaction data), and that it depends on
how the modules are defined. I think the authors could also discuss potential biases
here. Depending on the signal/noise ratio of a gene and the module size, how would
connectivity be affected? One would want to make sure that these factors are not
influential on the reported result.

Figure 2: Would it not be informative to add a
violin plot (similar to A and B) for dN/dS? Especially so, as lower conservation among
up-regulated genes is one of the paper's main points. But no information is given
regarding the magnitude of the effect. The authors could also plot expression versus
connectivity.

In the sixth paragraph of the Results section: There is little discussion on the GO
analysis. Does the UV response pathway have to do with sudden exposure to the sun? At
least would one not expect to see the same pathway up-regulated in foragers of other
taxa?

Please indicate the p-value cutoffs for the GO analysis. This is also found in the
Supplementary Material, but should be in the main text or Methods.

It would be helpful if the authors addressed the following:

What is the estimated genome size? What was the CEGMA assembly score for the de novo
genome assembly? What was the average coverage per sample for the genomic and
transcriptomic data?

The main conclusion that “genes unregulated in foragers and nurses were on
average less connected and more rapidly evolving” (ninth paragraph of the Results
section) relies heavily on the assumption that they are working with a high-quality
transcriptome and that their orthology assignments are correct.

How did they evaluate this? A table with summary statistics would be very useful. How
many transcripts had homology to the fire ant and/or the honey bee? How was the paralog
problem dealt with, particularly with respect to the molecular evolution analyses?

Similarly, for the network analyses: Were these co-expression networks calculated only
on significant transcripts or on all transcripts? How was a significant
“network” determined? Two of the modules had > 8000 transcripts in
each of them. Does that mean all 8000 transcripts show tightly-correlated expression
levels?

Finally, why didn't the authors include *Polistes* in their
comparative analyses? There are at least 2 studies on *Polistes*, both of
which are already cited in this manuscript. This seems like it would be another
independent data point worth discussing.

---

## [Author Response]

*While all three reviewers are enthusiastic about the study, several important
concerns were voiced by all of them. All reviewers pointed out that this paper does
not provide the necessary statistical details to be able to assess the quality of the
work. This is certainly the most important concern, as hypothesis
acceptance/rejection*—*which is the central message of the
study*—*fully depends on gene expression analysis and its
interpretation. The respective comments are combined below and it will be necessary
to address them before a final decision about acceptance can be reached*.

The statistical analysis for this project, as for most other bioinformatics projects, is
quite complex, and we chose to present the complete analytical pipeline and data as a
supplement to make it completely reproducible. We may have relied too much on our
supplement containing the complete analysis, omitting details from the main text. In
this revision we spent a lot of time addressing statistical questions and included the
relevant details in the main text. We also added several additional analyses.

*More specifically*:

*The authors investigate the “novel social genes” vs the
“gene toolkit” hypotheses. This is certainly interesting and
worthwhile. However, one general issue I have with this (and most other) studies on
the topic is that it is not clear what “exactly” constitutes evidence
for one hypothesis or the other*.

For example, the authors here state that the percentage of genes differentially
expressed in the same context in the two ants is small (∼3%). However, the
overlap is marginally significant (fifth paragraph of the Results section). So the
question is, what percent overlap exactly would constitute support for the toolkit
hypothesis? 50%? 25%? 10% 5%?

*The point is that it isn't clear how and when authors reject or fail to
reject the toolkit/novel hypotheses because there is never an explicit significance
or overlap threshold provided. The authors are not alone in having to deal with this
issue, and so I don't want to lay this completely at their feet, but it would
be nice if they stated explicitly what exactly would constitute evidence, or lack
thereof, for the toolkit hypothesis somewhere before they get to the
results*.

We agree completely. We struggled to come up with a satisfying approach to
quantitatively test these hypotheses, but finally came to the conclusion that, as
currently described in the literature, both the “genetic toolkit” and
“novel social genes” hypotheses are somewhat ambiguous, such that it is
not currently possible to really test these hypotheses. Similarly, a recent book chapter
by Adam [78] points out that the
concept of a developmental genetic toolkit is widely accepted, but it is also very
unclear what actually makes up the genetic toolkit, so that the hypothesis is very
difficult to actually test. Because these two hypotheses are entrenched in the
literature—and the idea of a genetic toolkit underlying social behavior is
frequently cited as *the* major finding of sociogenomic
research—we describe them in the Introduction to motivate our work. In the
Discussion, we explain how our results fit with these two hypotheses, but then we move
on to our main, and we believe, most exciting point: the genes we identified as being
associated with ant division of labor show different patterns of connectivity and
evolutionary conservation, and thus the genetic architecture of ant division of labor
includes both highly connected and conserved genes as well as more loosely-connected and
evolutionarily labile genes.

*Related to this issue, I wanted more detail about the analyses comparing
expression patterns between taxa. Did the authors just ask if the same genes were
differentially expressed between behavioral types in the two taxa to be viewed as
'consistent' with the toolkit model? Or did a particular gene have to
be differentially expressed in the same direction in both taxa? (The latter, I think,
although this wasn't clear.) And did a gene need to be significantly
differentially expressed to be counted? Or was it sufficient that it show the same
directionality in expression, regardless of significance? Directionality without
significance is as important as significance, given that the studies in the taxa had
different power, used different methods, etc. These are not trivial issues and they
may affect the outcome and interpretation of the results. I urge the authors to look
into this more closely*.

Following this suggestion we conducted the suggested analysis using the fire ant data.
When we look at just the direction of expression, or the fold-change in expression
across the genome, we find results very similar to those obtained when we only consider
differentially expressed genes. Whereas we previously found a 3.2-3.8 percent overlap in
differentially expressed genes between the fire ant and pharaoh ant data set, the
analysis based on the correlation of log fold change between nurse and forager samples
in the two species involves thousands of genes, but explains only 2% of the variance.
So, the results of the two analyses are consistent. This significant but seemingly low
overlap is also consistent with previous studies interpreted as supporting the genetic
toolkit hypothesis. For example, [81] found 18 brain-expressed genes associated with feeding behavior in both
the bumblebee *Bombus terrestris* and the paper wasp *Polistes
metricus*, which was more than expected by chance. Considering only genes
with identifiable homologs between the two study species and *A.
mellifera*, this corresponds to an overlap of 7.5% (18/239), but considering
all 2,563 identified *B. terrestris* feeding-associated genes, this
corresponds to an overlap of only 0.7% (18/2563).

*In the third paragraph: The authors state they are interested in the
“molecular mechanisms of social interactions (e.g. social signal production,
reception and response)”. They refer to genes related to social behavior
throughout the manuscript. But are their expression profiles indeed reflecting social
information producing/processing skills? Or morphological changes related to other
functions, such as exposure to exterior environments? They do not state they used
whole ants or just ant heads for transcriptome profiling, which is highly crucial for
interpreting the results (especially given that some other studies have used the
animal head)*.

We agree that these are very important points. We used whole ant bodies and we emphasize
this point more in the revised text. We also agree that the differentially expressed
genes we observed may be related to a wide range of physiological and behavioral
processes, and not just social signal production and response. The underlying
physiological processes are particularly intermeshed because by definition, age-based
division of labor in social insects means that the individuals are aging as they
transition from nursing to foraging tasks. We have clarified these issues in the revised
text.

In the third paragraph of the Results section: “Of these contrasts, only
foragers and nurses had significantly different gene expression patterns.”
This is not well explained. Four categories are compared and 2 of these are said to
have different expression. What is the reference here; are the other two categories
(grooming and trophallaxis) not different from these two? Or perhaps I am missing
something?

*This can be partly followed in the Supplementary Material, but should be
referred in the main text, e.g. it seems as if all samples were grouped together in
the DE analysis. The foragers and nurses were most different as they represent the
youngest and oldest. I would have stated this explicitly*.

There were four behavioral categories: foragers, nurses, and workers engaged in
trophallaxis and grooming. We wished to see which of the categories were
transcriptionally distinct from the others, and made comparisons of the focal category
vs all of the others. This section was indeed unclear and we re-wrote it with extra
detail.

*I am also concerned somewhat about the PCA in the Supplementary Material: There
seems to be two groups emerging, but this is likely technical (I would guess sample
processing dates). It might be difficult to control for this, but if possible, could
improve DE analysis significantly*.

All the samples were processed simultaneously and in the same facility. We cross-checked
the samples, and the two groups don’t correspond to sequencing lanes or to any
other discernable categories. The within-sample replicates, except for the loosely
defined behavioral category ‘trophallaixis’ are close to each other,
suggesting that the biological signal is strong in the data. Most importantly, the
separation between the clusters does not correlate with any biological signal detected
in the data. Finally, worker samples on the first day of eclosion (age 0) are an outlier
on this plot. As these workers are just beginning their transition to adult life, with
numerous physiological changes, such as cuticle hardening, it is logical to expect that
they would be separate from the others. If we discount these points, there is far less
evidence for the existence of separate groups in the scatter.

*In the third paragraph of the Results section: “There were 1217 forager-
and 1247 nurse-upregulated genes”. What was the p-value cutoff? How did the
authors control for multiple-testing? (This can also be followed in the Supplementary
Material, but should be referred in the main text*.*)*

We followed the standard practice of 0.05 cutoff after using FDR to adjust for multiple
comparisons. We now specify this in the new Statistical Analysis section of the
Methods.

*In the fourth paragraph of the Results section: “(…) it separated
workers into two distinct classes based on age” If I understand what was done,
I think the authors might be overinterpreting: the algorithm will separate the
profiles into 2 classes if k=2, and n classes if k=n. Thus, without
additional analysis I think one cannot decide on the existence of distinct classes.
The authors could consider applying some other test; e.g. check the slope of the
expression-age curve*.

We were interested in testing whether our temporal data could be classified into two
groups, corresponding to nurses and foragers, and, if so, where would the transition
point be. In the analysis, k=2 refers not to the number of classes, but to the
number of nearest neighbors considered by the k-nearest neighbor classifier to make the
distinction. We obtain basically the same result with k=2 and k=3, which
is within the range of the rule-of-thumb recommendations (references listed in KNN
section of the complete analysis). We also include a simple cluster diagram as Figure 1—figure supplement 1, to show that
the there is indeed a breakpoint before day 12. We have also clarified the logic behind
this analysis in the text.

*In the fifth and sixth paragraphs of the Results section: In the gene expression
conservation analysis, we are given no information how many genes are used in the
comparisons (i.e, the number of genes showing DE in both this and the other datasets,
as well as background genes). If the numbers are low, they could instead check the
effect size of orthologous genes identified as DE for honeybee, for example. Was the
honeybee data generated by Manfredini et al., 2014? If not, the authors should state
that*.

We added the numbers of genes, which were indeed low, and a parallel analysis using
effect sizes (log fold expression change) for fire ants, which produced the same result.
Unfortunately, for the honey bee study, only the list of differentially expressed genes
is publicly available, so that we could not perform a similar analysis for the honey bee
data set. The unavailability of previous data sets is one reason that we have strived to
make our full analysis available and completely transparent, as well as making all of
the data available.

*Most importantly, if the honeybee data was generated from the brain (as done by
quite a few studies) and the data in this study from the whole body, this could also
be a reason for finding limited overlap*.

*In comparisons with the Fisher's exact test, it would be useful to state
what the background is (non-DE genes, genes up-regulated in the other category, or
both?)*.

*The expression “whether genes differentially expressed in these
categories of workers were more likely conserved” is a bit confusing, as it
also implies sequence conservation, but I think the authors mean conservation with
respect to correlated changes*.

The honeybee data was indeed generated from the brain, and this is certainly a plausible
reason for the low observed overlap. We have added this very important point to our
discussion, qualifying our conclusions in this light. In the revised manuscript we focus
much less on the proportional overlap of the pharaoh ant and fire ant/honey bee data
sets, because of these methodological limitations of the comparisons, and as discussed
above, it is difficult to interpret what exactly these overlaps mean.

*In the seventh paragraph of the Results section: Connectivity—this could
be more explicitly defined, such as emphasizing that the prediction comes from
transcription data correlations (e.g. not protein-protein interaction data), and that
it depends on how the modules are defined. I think the authors could also discuss
potential biases here. Depending on the signal/noise ratio of a gene and the module
size, how would connectivity be affected? One would want to make sure that these
factors are not influential on the reported result*.

We added a more detailed description of WGCNA to the Methods section, including
citations to studies evaluating WGCNA performance. For our measure of connectivity, we
used the total connectivity of a gene, which is less sensitive to how modules are
defined, and most likely reflect the overall role of the gene, beyond the modules we
detect in our data set. Although the authors of the WGCNA package suggest that the
method can run on normalized count data, in the course of considering potential biases,
we found the gene lengths varied somewhat among behavioral categories. We then re-ran
the analysis using FPKM data, which are length-standardized. This analysis also captured
major network effects, but had a much better fit to the data. In particular, we had to
use a smaller soft thresholding level before an approximately scale-free topology of the
network was observed (a WGCNA requirement). We were also able to detect many more
modules at the same cutoff levels, suggesting greater network resolution.

The most obvious remaining bias from this sort of analysis is that genes with low
expression and low variability will not be detected as differentially expressed. Indeed,
this effect can clearly be seen in Figure 2,
with the average expression level of non-differentially expressed genes is lower.
However, contrary to what you would expect if the pattern was driven by this bias, the
nurse-upregulated genes show lower connectivity than non-differentially expressed genes
(Figure 2). We also explicitly control for
expression level by including expression level in our GLM analyses.

Figure 2*: Would it not
be informative to add a violin plot (similar to A and B) for dN/dS? Especially so, as
lower conservation among up-regulated genes is one of the paper's main points.
But no information is given regarding the magnitude of the effect. The authors could
also plot expression versus connectivity*.

We have added each of these suggested plots and have also added plots showing the
effects of connectivity and expression on whether genes had identifiable fire ant and
honey bee orthologs (Figure 3).

*In the sixth paragraph of the Results section: There is little discussion on the
GO analysis*. *Does the UV response pathway have to do with sudden
exposure to the sun? At least would one not expect to see the same pathway
up-regulated in foragers of other taxa?*

*Please indicate the p-value cutoffs for the GO analysis. This is also found in
the Supplementary Material, but should be in the main text or Methods*.

We have added further discussion of the GO analysis and include the full set of GO terms
enriched for both the forager- and nurse-upregulated genes as well as each module
separately. We do not dwell on GO terms since approximately 50% of all genes in our
analysis do not even have identifiable orthologs, so we cannot generally be confident
about inferring function of differentially-expressed genes or gene modules.

*It would be helpful if the authors addressed the following*:

What is the estimated genome size? What was the CEGMA assembly score for the de
novo genome assembly? What was the average coverage per sample for the genomic and
transcriptomic data?

As we mention in response to Reviewer 1, a genome project was not the goal of this
study. That being said, as the quality of the reference genome assembly is important to
this study, we now include quality control statistics in the Results, such as coverage
statistics and CEGMA, as requested. Unfortunately, there is no independently estimated
size of the *M. pharaonis* genome.

*The main conclusion that “genes unregulated in foragers and nurses were
on average less connected and more rapidly evolving” (ninth paragraph of the
Results section) relies heavily on the assumption that they are working with a
high-quality transcriptome and that their orthology assignments are
correct*.

How did they evaluate this? A table with summary statistics would be very
useful. How many transcripts had homology to the fire ant and/or the honey bee? How
was the paralog problem dealt with, particularly with respect to the molecular
evolution analyses?

As with all studies involving gene orthology, particularly with-non models species,
there is no genome-wide gold standard that allows the performance of a method to be
evaluated. Because the data set is based on transcriptomic data, there will necessarily
be false negatives associated with poorly expressed transcripts. That being said, our
choice of reciprocal best hit is well justified, based on comparisons of various
available methods, as it has high specificity, though at the cost of sensitivity, as it
ignores paralogous relationships for genes that have been duplicated in one of the
lineages (12).

Another important consideration is that we used an independently assembled genome to
estimate evolutionary rates. Consequently, our measurements of nucleotide-level changes
were not dependent on the quality of the transcriptome, except that some regions of the
gene may be missing due to poor transcript coverage. So, transcript quality should not
bias the evolutionary rate estimates.

*Similarly, for the network analyses: Were these co-expression networks
calculated only on significant transcripts or on all transcripts? How was a
significant “network” determined? Two of the modules had > 8000
transcripts in each of them*. *Does that mean all 8000 transcripts
show tightly-correlated expression levels?*

The networks were calculated using all transcripts and all samples. There are no
measures of module significance per se, and the actual module structure may change if
different cutoffs are chosen in the analysis (see [Supplementary-material SD2-data]).
Indeed, in the current analysis, the number of modules is greater than the original
analysis. The effect of cutoff on the number of modules can be seen in the [Supplementary-material SD2-data]. As
described above, we use WGNA to estimate total network connectivity (i.e.
“kTotal” in WGCNA) for each node (i.e. gene), and present the module trait
correlation heat map as a low-dimensional visualization of the underlying
transcriptional changes.

As we discuss in the manuscript, the true measure of a module’s validity may be
its repeatability across different studies within and between lineages. Thus, it will be
important to cross-check the modules found in our data with future studies conducted
using comparable methodology. This is another reason we are including the complete data
with our analysis.

*Finally, why didn't the authors include* Polistes *in
their comparative analyses? There are at least 2 studies on*
Polistes*, both of which are already cited in this manuscript. This seems like
it would be another independent data point worth discussing*.

The *Polistes* paper wasp studies focus on reproductive division of labor
between queens and workers and do not include data on worker age polyethism, and are
thus not directly comparable with our study. Even though age polyethism is thought to be
widespread in social insects, the transcriptomic basis of age polyethism has only been
studied in the honey bee *Apis mellifera* and the fire ant
*Solenopsis invicta*, with the fire ant study only comparing the two
extremes of workers inside the nest and workers outside the nest (43).